# Search for lepton-flavor-violating decays of the tau lepton at a future muon collider

Gholamhossein Haghighat[1], Mojtaba Mohammadi Najafabadi[1]

**1** School of Particles and Accelerators, Institute for Research in Fundamental Sciences (IPM),
P.O. Box 19395-5531, Tehran, Iran
* h.haghighat@ipm.ir

April 12, 2022

*16th International Workshop on Tau Lepton Physics (TAU2021),
September 27 – October 1, 2021*

## Abstract

**Tau leptons can have lepton-flavor-violating (LFV) couplings to a muon or an electron and an Axion-Like Particle (ALP). ALPs are pseudo Nambu-Goldstone bosons associated with spontaneously broken global U(1) symmetries. LFV ALPs have been of a great interest in the last several decades as they can address some of the SM long-lasting problems. Assuming a future muon collider proposed by the Muon Accelerator Program (MAP), we search for LFV decays $\tau \to \ell a$ ($\ell = e, \mu$) of one of the tau leptons produced in the muon-anti muon annihilation. The ALP mass is assumed to be in the range 100 eV to 1 MeV and three different chiral structures are considered for the LFV coupling. Using a multivariate technique and performing a parameterized simulation based on the ideal target performance, we obtain expected 95% confidence level upper limits on the LFV couplings tau-electron-ALP and tau-muon-ALP. Limits are computed assuming the center-of-mass energies of 126, 350 and 1500 GeV which the future muon collider is supposed to operate at. We study the two cases of unpolarized and polarized muon beams and show that taking advantage of tau polarization-induced effects, the main background $\tau \to e/\mu + \nu\bar{\nu}$ can be significantly reduced. Results indicate that current limits on the LFV couplings can be improved by roughly one order of magnitude using the present analysis.**

# 1 Introduction

The neutrino oscillations [1] verified by experimental data suggest the possibility of charged lepton-flavor-violating (LFV) decays. These decays are strongly suppressed in the Standard Model (SM) providing a strong motivation for searching for beyond SM (BSM) LFV processes. Axion-Like Particles (ALPs), on the other hand, are pseudo Nambu-Goldstone bosons associated with spontaneously broken U(1) symmetries [2] and can have flavor-violating couplings to the SM charged leptons. Such particles have found numerous applications in many areas of physics as they can address some of the SM problems such as the strong CP problem, baryon asymmetry problem, Dark Matter, etc. The parameter space of LFV ALPs has been probed by many experiments [3] in the recent decades. Collider searches for the tau decay $\tau \to e/\mu + a$, where $a$ denotes an ALP, have constrained the ALP-tau-electron and ALP-tau-muon couplings [4]. ALPs with $\leq 1$ MeV masses decay predominantly to photons. However, they leave the detector before the decay and thus their existence can be revealed by missing energy signals. In such searches, the signal is mainly overwhelmed by the decay $\tau \to e/\mu + \nu\bar{\nu}$. In this work, we provide a procedure for separating the signal from this background using tau polarization-induced effects and show that this method can improve the obtained sensitivity significantly. Assuming a future muon collider proposed by the Muon Accelerator Program (MAP) [5], we study both the tau decay final states $e^{\pm} + a$ and $\mu^{\pm} + a$ and provide the expected limits on the couplings ALP-tau-muon and ALP-tau-electron for ALPs with $m_a \leq 1$ MeV.

# 2 ALP production in LFV tau decays

The LFV coupling of the ALP to the SM charged leptons can be described by the effective Lagrangian [3]

$$\mathcal{L}_{\text{eff}} = \sum_{i \neq j} \frac{\partial_\mu a}{2f_a} \bar{\ell}_i \gamma^\mu (c^V_{\ell_i \ell_j} + c^A_{\ell_i \ell_j} \gamma_5)\ell_j, \tag{1}$$

where $c^{V,A}_{\ell_i \ell_j}$ are hermitian matrices, $f_a$ is the symmetry breaking scale of the model and $\ell_i = e, \mu, \tau$. We assume the chiral structures V+A, V−A and V/A. The V+A and V−A cases respectively correspond to the $c_{\ell_i \ell_j} \equiv c^V_{\ell_i \ell_j} = c^A_{\ell_i \ell_j}$ and $c_{\ell_i \ell_j} \equiv c^V_{\ell_i \ell_j} = -c^A_{\ell_i \ell_j}$ conditions. The V/A case assumes that one of the $c^{V,A}_{\ell_i \ell_j}$ couplings vanishes, i.e. either $c_{\ell_i \ell_j} \equiv c^V_{\ell_i \ell_j} \neq 0, c^A_{\ell_i \ell_j} = 0$ or $c_{\ell_i \ell_j} \equiv c^A_{\ell_i \ell_j} \neq 0, c^V_{\ell_i \ell_j} = 0$. According to the above Lagrangian, the ALP can be produced in the tau decays $\tau^{\pm} \to e^{\pm} a$ and $\tau^{\pm} \to \mu^{\pm} a$. The ALP mass is assumed to be in the range $m_a \leq 1$ MeV in this study. Such light ALPs can only decay into a pair of photons and have a decay length many orders of magnitude larger than the typical size of a detector. The majority of them, therefore, manifest themselves as missing energy.

   We assume the process $\mu^- \mu^+ \to \tau^- \tau^+$ with the subsequent decay of one of the tau leptons into $e/\mu + a$. The remaining tau lepton experiences a SM decay. A future muon collider proposed

by the Muon Accelerator Program (MAP) operating at the center-of-mass energies 126, 350 and 1500 GeV is assumed. In the limit $m_a \ll m_\tau$, the final lepton in the decay $\tau \to \ell\, a$ ($\ell = e, \mu$) has an energy of $E_\ell \simeq m_\tau/2$. Neglecting the final state lepton mass in this decay, the rest frame differential decay width for the three assumed chiral structures are found to be [3]

$$\frac{d\Gamma(\tau^\pm \to \ell^\pm a)}{d\cos\theta} = \frac{m_\tau^3}{128\pi f_a^2}\left(1 - \frac{m_a^2}{m_\tau^2}\right)^2 \times \begin{cases} 2c_{\tau\ell}^2\,(1 \mp \mathcal{P}_\tau \cos\theta\,) & \text{V+A} \\ 2c_{\tau\ell}^2\,(1 \pm \mathcal{P}_\tau \cos\theta\,) & \text{V$-$A} \\ c_{\tau\ell}^2 & \text{V/A} \end{cases}, \qquad (2)$$

where $\theta$ is the angle between the polarization vector of the decaying tau and the direction of the final state lepton and $\mathcal{P}_\tau$ is the degree of polarization of tau leptons. As seen, the angular distribution of the final lepton depends on the assumed chiral structure. Despite the final lepton in the LFV tau decay which is monoenergetic regardless of its direction, in the SM decay $\tau \to e/\mu + \nu\bar{\nu}$, the energy of the final lepton depends on its angular orientation. To be more specific, final leptons $\ell^-$ ($\ell^+$) emitted forwardly (backwardly) with respect to the polarization of the decaying tau are less energetic than those emitted in other directions and are mostly separated from the leptons resulting from the LFV tau decay. The difference between the polarization-induced effects in these processes is utilized in this study to separate the LFV signal from the main SM background. In addition to the main SM background, other background processes, i.e. the production of $W^- W^+$, $t\bar{t}$, $ZZ$, $Z\gamma$, $hZ$, $q\bar{q}$ ($q = u, d, c, s, b$) and $e^- e^+/\mu^- \mu^+$ are also considered in this analysis. At high enough center-of-mass energies, the SM vector boson fusion (VBF) processes become important and should be considered in any analysis. However, at the (relatively low) center-of-mass energies assumed in this study, the cross section of the VBF processes is so small that the contribution of these processes to the background can be safely neglected [6].

## 3   Event generation and analysis

Generation of events is performed using the packages `FeynRules` [7], `MadGraph5_aMC@NLO` [8], `Pythia 8.2.43` [9], `Delphes 3.4.2` [10] with the muon collider delphes card [1] and `FastJet` `3.3.2` [11]. The center-of-mass energies 126, 350 and 1500 GeV corresponding to the integrated luminosities of 2.5, 189.2 and 394.2 fb$^{-1}$ are assumed and the event generation is performed separately for the cases of unpolarized muon beams and polarized muon beams with $+0.8$ ($-0.8$) polarization for the $\mu^-$ ($\mu^+$) beam. The ALP mass is assumed to vary in the range 100 eV to 1 MeV. In the event generation, $f_a$ is set to 10 TeV. The two cases of $c_{\tau\mu} \neq 0$ and $c_{\tau e} \neq 0$ are analyzed independently. Appropriate event selection criteria regarding the number and type of the reconstructed objects in the events are performed. $\tau$-tagged jets should satisfy the kinematical conditions $p_T > 30$ GeV and $|\eta| < 2.5$. Moreover, they should include exactly three charged hadrons, and at most one photon. Isolated muons, electrons and photons are required to pass the conditions $p_T > 10$ GeV and $|\eta| < 2.5$. Events should either have exactly one electron-muon pair or exactly one lepton (electron or muon) along with one $\tau$-tagged jet. Using the momentum direction of the tau leptons, the space is divided into two hemispheres, one on the side of the tau lepton undergoing the LFV decay and one on the side of the tau lepton undergoing the SM decay. There can be at most one photon on either side. Events passing the selection cuts are analyzed to compute a proper set of discriminating variables. A multivariate technique using the Boosted

---

[1]https://github.com/delphes/delphes/blob/master/cards/delphes_card_MuonColliderDet.tcl

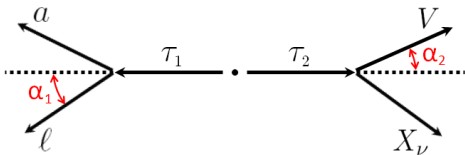

Figure 1: The decaying tau leptons in the ALP production process. $\tau_1$ and $\tau_2$ respectively undergo the LFV and SM decays. $V$ is either a lepton $(e, \mu)$ or a system of hadrons identified as a $\tau$-jet, and $X_\nu$ is a system of neutrinos.

Decision Trees (BDT) algorithm [12] is deployed to better discriminate between the signal and background. The BDT output is used to constrain the ratios $c_{\tau e}/f_a$ and $c_{\tau \mu}/f_a$.

The cartoon in Fig. 1 shows the decaying tau leptons in the ALP production process. The momenta of the decaying tau leptons cannot be exactly determined because of the invisible ALP and neutrinos. Using the conservation of momentum, one finds the system of equations

$$\hat{v}_1 \cdot \vec{p}_\ell = \frac{2E_{\tau_1}E_\ell + m_a^2 - m_\tau^2 - m_\ell^2}{2\sqrt{E_{\tau_1}^2 - m_\tau^2}}, \qquad \hat{v}_1 \cdot \vec{p}_V = -\frac{2E_{\tau_2}E_V - m_\tau^2 - m_V^2}{2\sqrt{E_{\tau_2}^2 - m_\tau^2}}, \qquad |\hat{v}_1| = 1, \qquad (3)$$

where the unit vector $\hat{v}_1$ denotes the momentum direction of $\tau_1$. $E_{\tau_1}$ and $E_{\tau_2}$ can be estimated, ignoring the radiative emissions and using the conservation of energy, to be half of the center-of-mass energy of the experiment, i.e. $E_{\tau_1} = E_{\tau_2} = \sqrt{s}/2$. This system of equations has two solutions for $\hat{v}_1$. We take the average of the solutions as $\hat{v}_1$ and reconstruct the momenta of the decaying tau leptons $\vec{p}_{\tau_1}$ and $\vec{p}_{\tau_2}$, using the obtained momentum direction. Furthermore, we reconstruct the momentum of the invisible ALP $(E_a, \vec{p}_a)$.

The discriminating variables used by the BDT algorithm are provided below. The frame in which a variable is measured is the laboratory frame unless stated otherwise.

- Missing transverse energy $\not{E}_T$.

- Transverse momenta and pseudorapidities of the isolated lepton $\ell$ and the $V$ system.

- Invariant mass of the the ALP and the isolated lepton $\ell$.

- Angle between the momentum of the $V$ system and the momentum of the lepton $\ell$.

- Angle between the momenta of the lepton $\ell$ and the ALP in the $\tau_1$ rest frame.

- Angle between the momentum of $\tau_1$ in the laboratory frame and the momentum of the lepton $\ell$ in the $\tau_1$ rest frame.

- Energy fraction of the lepton $\ell$ in the $\tau_1$ rest frame, $x_\ell^{\tau_1 \text{RF}} = 2E_\ell^{\tau_1 \text{RF}}/m_\tau$.

A comparison shows that the signal-background discrimination using the above variables is significantly improved in the polarized muon beams case compared with the unpolarized case.

## 4 Prospects for limits on the LFV tau couplings

Using the BDT output, we obtain expected 95% confidence level (CL) limits on the couplings $c_{\tau e}/f_a$ and $c_{\tau \mu}/f_a$. We assume a 10% systematic uncertainty on the event selection efficiency of

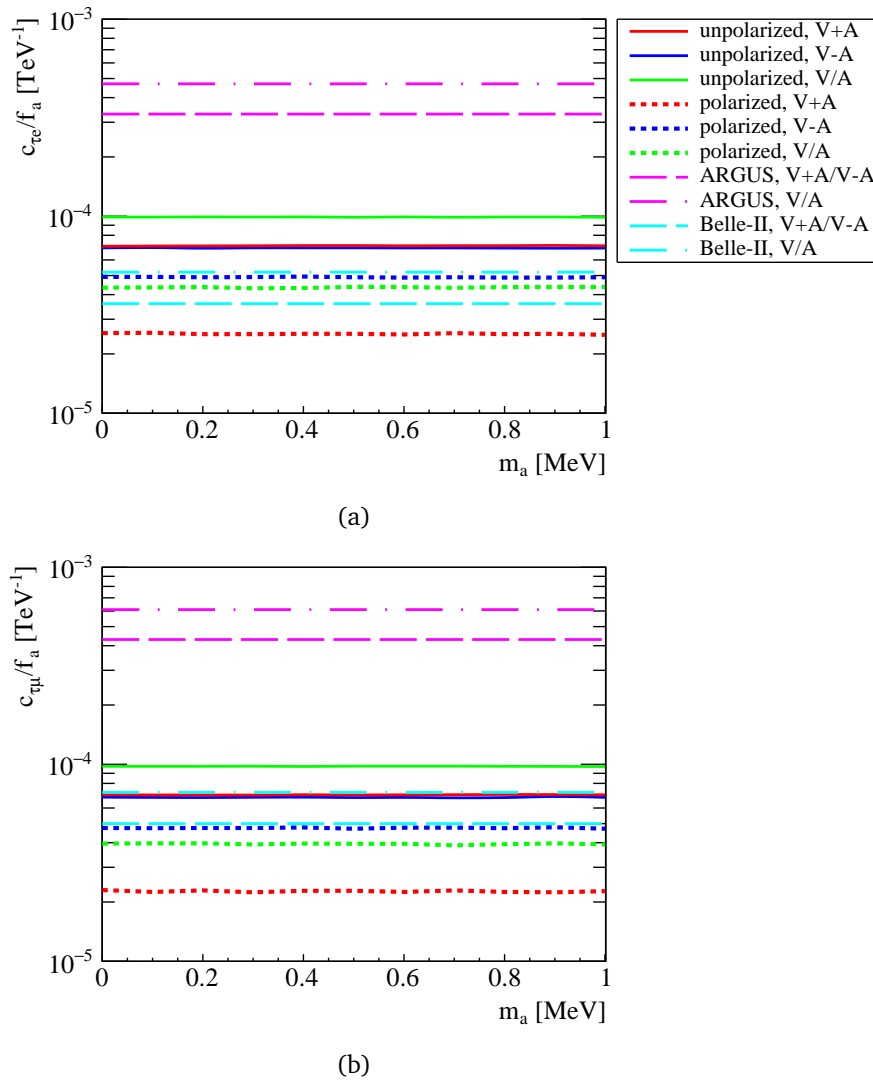

Figure 2: Obtained 95% CL expected limits at $\sqrt{s} = 350$ GeV, current limits and future prospects of the limits on a) $c_{\tau e}/f_a$ and b) $c_{\tau \mu}/f_a$ against the ALP mass. The green, blue and red solid (dotted) lines respectively show the expected limits in the V/A, V−A and V+A cases with unpolarized (polarized) beams. The pink dashed (dash-dotted) line shows the present limit in the V+A/V−A (V/A) case obtained in ARGUS searches. The cyan dashed (dash-dotted) line shows the Belle-II future prospects for the V+A/V−A (V/A) structure assuming an integrated luminosity of 50 $ab^{-1}$.

each process. Results indicate that the most stringent limits belong to the center-of-mass energy of 350 GeV, and that different ALP mass scenarios below 1 MeV result in similar limits. Fig. 2 shows the limits obtained at $\sqrt{s} = 350$ GeV against the ALP mass for the three assumed chiral structures and the two muon polarization cases. Also shown are the present limits derived from searches at the ARGUS experiment [4] as well as future prospects at Belle-II [13]. It is seen that both the obtained unpolarized and polarized limits are significantly stronger than the present limits. Assuming an ALP mass of 1 MeV, the most stringent limits on $c_{\tau e}/f_a$ ($c_{\tau \mu}/f_a$) in the V+A, V−A and

V/A cases belong to the polarized muon beams case and are respectively $2.48 \times 10^{-5}$ ($2.27 \times 10^{-5}$), $4.90 \times 10^{-5}$ ($4.68 \times 10^{-5}$) and $4.37 \times 10^{-5}$ ($3.92 \times 10^{-5}$) TeV$^{-1}$. Our recast of the limits derived from ARGUS searches in the V+A/V−A (V/A) case are $c_{\tau e}/f_a < 3.3 \times 10^{-4}$ ($c_{\tau e}/f_a < 4.7 \times 10^{-4}$) and $c_{\tau \mu}/f_a < 4.3 \times 10^{-4}$ ($c_{\tau \mu}/f_a < 6.1 \times 10^{-4}$) TeV$^{-1}$. A comparison shows that the presented analysis can improve the current limits by roughly one order of magnitude. Results also indicate that the obtained limits in the unpolarized case are slightly weaker than the prospects at Belle-II. In the polarized case, however, the obtained limits for the V+A and V/A chiral structures are slightly stronger than the prospects at Belle-II.

## 5   Conclusion

We studied the LFV decays of the tau lepton into an electron (or a muon) and an ALP assuming a future muon collider operating at $\sqrt{s} = 126$, 350 and 1500 GeV. The chiral structures V+A, V−A and V/A for the LFV tau decay, and the two cases of unpolarized and polarized muon beams were studied separately and expected 95% CL limits on the LFV couplings $c_{\tau e}/f_a$ and $c_{\tau \mu}/f_a$ for ALP masses $m_a \leq 1$ MeV were obtained. The most stringent limits were obtained at the center-of-mass energy of 350 GeV. It is seen that, if producing the polarized muon beams assumed in this work becomes possible, the present experimental limits derived from ARGUS searches can be improved by roughly one order of magnitude with the help of this analysis. The improvement in the experimental limits using the present analysis with unpolarized muon beams can be about half an order of magnitude. A comparison shows that the limits obtained in the polarized case assuming the V+A and V/A chiral structures are slightly stronger than the prospects at Belle-II assuming an integrated luminosity of 50 $ab^{-1}$. Other obtained polarized and unpolarized limits are, however, slightly weaker than the prospects at Belle-II. It is seen that the limits get worse as the center-of-mass energy increases from 350 GeV to 1.5 TeV. Both the signal and background cross sections decrease as the energy grows. The reduction in the background, however, doesn't compensate for the reduction in the signal giving rise to looser constraints. At high enough energies, the vector boson fusion (VBF) processes become important. Such processes may further degrade the limits at high energies since they can contribute to the background.

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
