# Peer review of "Search for lepton-flavor-violating decays of the tau lepton at a future muon collider"

_SciPost Physics Proceedings_

## Round 1 · Referee Report · Anonymous · 2022-4-20

Report

The authors have addressed my suggestions for minor changes. I recommend this article for publication.

---

## Round 1 · Referee Report · Anonymous · 2022-4-25

Report

I'd like to thank the authors for addressing my comments.
I am happy with the revised version of the manuscript and I would recommend it for publication.

---

## Round 1 · Author Response

We would like to thank the referees for reviewing our manuscript. We are especially grateful for their insightful comments and suggestions. We have revised our manuscript in response to the comments and hope that this improved manuscript is acceptable for publication.

---

## Round 1 · List of Changes

Report of Referee (1):

Comment 1 — In the abstract and in the main text the authors claim that their analysis shows an improvement of the sensitivity of an order of magnitude w.r.t. current experimental bounds (coming from the ARGUS experiment). Technically, this cannot be seen in the plots of Figure 2, as the scale on the y-axis contains only one point. A few more values on the scale would help reading the plots.

Changes in manuscript: We modified the plots of fig. 2 so that the y-axis includes more values.

Comment 2 — Regarding the projected limits from Belle-II in comparison to the hypothetical muon collider, I believe that the statement by the authors is a bit too strong, and should be phrased in a clearer way. Indeed the plots show that, especially for unpolarised muon beams, the expected sensitivity from Belle-II is the same, if not better, while the muon collider seems to be doing only slightly better with polarised beams, and only in the case of V+A couplings.

Changes in manuscript: We have modified the last sentence in section 4, “Furthermore, a comparison with the future prospects at Belle-II shows that the limits obtained in this study are sensibly stronger.”, to “Results also indicate that the obtained limits in the unpolarized case are slightly weaker than the prospects at Belle-II. In the polarized case, however, the obtained limits for the V+A and V/A chiral structures are slightly stronger than the prospects at Belle-II.” Furthermore, we have modified the last three sentences in section 5, “It was seen that the present experimental limits derived from ARGUS searches can be improved by roughly one order of magnitude with the help of the present analysis. Furthermore, the obtained limits are sensibly stronger than the Belle-II future prospects assuming an integrated luminosity of 50 $ab^{−1}$ . It was also seen that the limits obtained assuming unpolarized muon beams are significantly stronger than the present limits.”, to “It is seen that, if producing the polarized muon beams assumed in this work becomes possible, the present experimental limits derived from ARGUS searches can be improved by roughly one order of magnitude with the help of this analysis. The improvement in the experimental limits using the present analysis with unpolarized muon beams can be about half an order of magnitude. A comparison shows that the limits obtained in the polarized case assuming the V+A and V/A chiral structures are slightly stronger than the prospects at Belle-II assuming an integrated luminosity of 50 $ab^{−1}$. Other obtained polarized and unpolarized limits are, however, slightly weaker than the prospects at Belle-II.”

Report of Referee (2):

Comment 1 — Does the event generation include VBF processes? These can become quite important especially in the highest centre-of-mass energy scenarios.

Changes in manuscript: We have added the following text at the end of section 2: “At high enough center-of-mass energies, the SM vector boson fusion (VBF) processes become important and should be considered in any analysis. However, at the (relatively low) center-of-mass energies assumed in this study, the cross section of the VBF processes is so small that the contribution of these processes to the background can be safely neglected [1].”

Comment 2 — The current plans for a muon collider start with a “low energy” stage with a centre-of-mass energy of 3 TeV, and plan to go higher. I believe it would be important to state/or show how the sensitivity to the LFV couplings is expected to change in these scenarios (for example comparing the 350 GeV results with those obtained at 1.5 TeV).

Changes in manuscript: We added the following text at the end of section 5: “It is seen that the limits get worse as the center-of-mass energy increases from 350 GeV to 1.5 TeV. Both the signal and background cross sections decrease as the energy grows. The reduction in the background, however, doesn’t compensate for the reduction in the signal giving rise to looser constraints. At high enough energies, the vector boson fusion (VBF) processes become important. Such processes
may further degrade the limits at high energies since they can contribute to the background.”

Comment 3 — The abstract mentions the use of a realistic detector simulation. I suggest to rephrase this to something like “a parameterized simulation based on the ideal target performance”

Changes in manuscript: We replaced “a realistic detector simulation” in the abstract with “a parameterized simulation based on the ideal target performance”.

Comment 4 — If possible, I think that giving a minimal description of (or reference for) the event selection criteria would be needed to eventually reproduce this study.

Changes in manuscript: We have added he following text in the middle of the first paragraph in section 3: “$\tau$-tagged jets should satisfy the kinematical conditions $p_T > 30$ GeV and $|\eta| < 2.5$. Moreover, they should include exactly three charged hadrons, and at most one photon. Isolated muons, electrons and photons are required to pass the conditions $p_T > 10$ GeV and $|\eta| < 2.5$. Events should either have exactly one electron-muon pair or exactly one lepton (electron or muon) along with one $\tau$-tagged jet. Using the momentum direction of the tau leptons, the space is divided into two hemispheres, one on the side of the tau lepton undergoing the LFV decay and one on the side of the tau lepton undergoing the SM decay. There can be at most one photon on either side.”

Comment 5 — Section 3 mentions $c_{\tau e}$ and $c_{\tau\mu}$ , while the results are presented in as the ratio of these couplings with f a . It would be good to harmonise the discussion and explicitly state which value of f a was assumed in the generation/interpretation.

Changes in manuscript: We have modified the last sentence of the first paragraph in section 3, “The BDT output is used to constrain the tau LFV couplings $c_{\tau e}$ and $c_{\tau\mu}$.”, to “The BDT output is used to constrain the ratios $c_{\tau e}/f_a$ and $c_{\tau\mu}/f_a$.” Furthermore, we have added the following text in the middle of the same paragraph: “In the event generation, $f_a$ is set to 10 TeV.”

Comment 6 — The concluding statement is somewhat strong, given that obtaining polarized muon beams would be highly unlikely. The authors might want to rephrase their closing statements to make them softer.

Changes in manuscript: We have modified the last three sentences in section 5, “It was seen that the present experimental limits derived from ARGUS searches can be improved by roughly one order of magnitude with the help of the present analysis. Furthermore, the obtained limits are sensibly stronger than the Belle-II future prospects assuming an integrated luminosity of 50 $ab^{−1}$ . It was also seen that the limits obtained assuming unpolarized muon beams are significantly stronger than the present limits.”, to “It is seen that, if producing the polarized muon beams assumed in this work becomes possible, the present experimental limits derived from ARGUS searches can be improved by roughly one order of magnitude with the help of this analysis. The improvement in the experimental limits using the present analysis with unpolarized muon beams can be about half an order of magnitude. A comparison shows that the limits obtained in the polarized case assuming the V+A and V/A chiral structures are slightly stronger than the prospects at Belle-II assuming
an integrated luminosity of 50 $ab^{−1}$ . Other obtained polarized and unpolarized limits are, however, slightly weaker than the prospects at Belle-II.”

References
[1] A. Costantini, F. De Lillo, F. Maltoni, L. Mantani, O. Mattelaer, R. Ruiz and X. Zhao, JHEP 09, 080 (2020) doi:10.1007/JHEP09(2020)080 [arXiv:2005.10289 [hep-ph]].

---

## Editorial Decision

accepted_in_target_journal